# Feasibility, acceptability, and safety of a novel device for self-collecting capillary blood samples in clinical trials in the context of the pandemic and beyond

Harika Dasari[1], Anna Smyrnova[1], Jing Leng[1], Francine M. Ducharme [1,2,3]*

1 Clinical Research and Knowledge Transfer Unit on Childhood Asthma, Centre Hospitalier Universitaire Sainte-Justine, Montreal, Quebec, Canada, 2 Department of Pediatrics, University of Montréal, Quebec, Canada, 3 Department of Social and Preventive Medicine, School of Public Health, University of Montreal, Québec, Canada

* Francine.m.ducharme@umontreal.ca

## Abstract

### Background

Home blood self-collection devices can enable remote monitoring, but their implementation requires validation. Our objectives were to explore (i) the impact of sampling sites and topical analgesia on capillary blood volume and pain perception and (ii) the safety, acceptability, and failure of capillary self-collection among adults and children using the Tasso-SST device.

### Methods

We conducted a two-phase study. The *investigational phase* consisted of two on-site cross-sectional studies in healthy adult participants ($\geq$ 12 years) and children (1–17 years) with their accompanying parent. Adults received 4 capillary samplings, where puncture sites and topical analgesia were randomized in a factorial design, and a venipuncture; children (and one parent) had one capillary sampling. The two co-primary outcomes were blood volume and pain. The *implementation phase* was conducted in two multicentre trials in participants choosing remote visits; blood volume, collection failure, adverse events, and satisfaction were documented.

### Results

In the *investigational phase*, 90 participants and 9 children with 7 parents were enrolled; 15 adults and 2 preschoolers participated in the *implementation phase*. In the adult investigational study, the device collected a median (25%, 75%) of 450 (250, 550) μl of blood with no significant difference between the puncture site, topical analgesia, and its interaction. Using topical analgesia reduced pain perception by 0.61 (95% CI: 0.97, 0.24; *P* <0.01) points on the 11-point scale; the pain reduction varied by puncture site, with the lower back showing the most significant decrease. Overall, combining all studies and phases, the median

**Data Availability Statement:** The dissemination of data to the public is precluded due to an omission pertaining to the inclusion of a data-sharing

provision within the consent forms, as mandated by the Research Ethics Board. Anonymized Data are available from the CHU Sainte Justine Institutional Data Access / Ethics Committee (contact via ethique.hsj@ssss.gouv.qc.ca) for researchers who meet the criteria for access to confidential data.

**Funding:** This work was funded by a grant awarded to FMD through a peer-reviewed process of the COVID-19 May 2020 Rapid Response Funding Opportunity by the Canadian Institute of Health Research [grant no. 447317]. The funders had no role in study design, data collection and analysis, decision to publish, or preparation of the manuscript.

**Competing interests:** The authors have read the journal's policy and have the following competing interests: Francine M. Ducharme has received unrestricted research funds from AstraZeneca, Covis Pharma, GlaxoSmithKline, Merck Canada, Novartis, Teva, Trudell Medical; GlaxoSmithKline and MEDteq in partnership with Thorasys Inc., as well as honorariums for consultancy work from AstraZeneca, Covis Pharma, Sanofi, Teva, and Thorasys Inc. and honorariums as an invited speaker from Covis Pharma, Jean-Coutu Pharmacy and Brunet Pharmacy. There are no patents, products in development or marketed products associated with this research to declare. This does not alter our adherence to PLOS ONE policies on sharing data and materials.

volume collected was 425 (250, 500) µl, and the device failure rate was 4.4%; minor adverse effects were reported in 8.9% of the participants, all were willing to use the device again.

## Conclusion

Capillary blood self-collection, yielding slightly less than 500 µl, proves to be a safe and relatively painless method for adults and children, with high satisfaction and low failure rates. The puncture site and topical analgesia do not affect blood volume, but topical analgesia on the lower back could reduce pain.

## Introduction

During the SARS-CoV-2 (COVID-19) pandemic, investigators sought alternative methods to venous phlebotomy to facilitate the pursuit of pediatric and adult clinical trials [1]. Several home dry-blood collection kits were available, but clinical laboratories generally could not rapidly develop dry blood extraction and analytic techniques during this period. A home blood collection kit that collects and transports wet blood to designated laboratories offered an attractive alternative. Although pediatric laboratories have developed innovative techniques for small capillary volumes, such a device must still meet the minimum volume required for a given (or a series of) analytic test(s). Our challenge was to identify a capillary sampling device that was easy to use at home, collected sufficient blood volume, and was both applicable and safe in children and adults. Pediatric use entailed three additional considerations: pain, alternative puncture site to the upper arm, as well as the potential need for, and impact of, topical anesthetic on pain and blood volume.

Multiple remote capillary blood collection kits were available in 2020, including the TAP Blood Collection Device (YourBio Health, Medford, USA), the Loop One device (Loop Medical, Lausanne, Switzerland), and the Tasso-SST device (Tasso, Inc., Seattle, USA). The TAP device had been cleared by the Food and Drug Administration (FDA) and Conformité Européene (CE) in Europe for use in adults 21 years and over; it reported an average collected blood volume of 100 µl [2–4]. The Loop One device could reach up to 1 ml but had not yet received regulatory approval [5]. Only the Tasso-SST device was approved for investigational use in individuals 3 months and older by the FDA and, for specific studies, by Health Canada. The manufacturer's reported average volume per device was 250 µl of capillary blood, [6] but it could perhaps be increased with different measures to increase capillary flow [7]. We sought a device to collect a minimal total volume of 600 µl and 400 µl for adult and pediatric trials, respectively [4, 8]. After careful consideration, we selected the Tasso-SST device due to its Health Canada approval, average reported volume, and ease of use, understanding that several devices would likely be needed to reach our target blood volume. Whereas the upper arm was recommended for adults, [4, 9] a focus group of pediatricians and parents, recommended the lower back for use in children under 4 years to reduce the risk of removal by the child (Erwin Berthier, personal communication, May 2020). As the neonatal heel prick test is considered more painful than venipuncture, [10] we sought but could not find published data on the relative discomfort of capillary sampling on the lower back or upper arm vs. venipuncture. While a topical anesthetic cream reduces the discomfort in pediatric venipuncture, [11] it was unclear whether it would be indicated and/or effective for capillary sampling. If used, vasoconstriction caused by the topical anesthetic could reduce collected volume [12].

The overarching objectives were to examine the blood volume and pain of capillary self-collection using the Tasso-SST device in a variety of settings and populations; secondary outcomes included safety profile, appreciation of the training offered, acceptability, and pragmatic issues (e.g., failure rate). But first, we wished to explore the impact of sampling sites and topical analgesia on capillary blood volume and pain in adolescents and adults.

## Methods

We conducted a two-phase study, including an investigational and an implementation phase, testing blood sampling with the Tasso-SST device [13–15]. The investigational phase consisted of two sequential cross-sectional studies conducted on-site at the Sainte Justine University Health Centre (SJUHC), Montréal, Canada, first in adults (September 23, 2020 to 12 May 2021) and then in children-parent dyads (June 29 to July 6, 2021). The adult cross-sectional study used a factorial design to test the impact of two factors (puncture site and topical analgesia) on capillary blood volume and pain as co-primary outcomes, using four devices. In the pediatric cross-sectional study, capillary blood volume and pain were measured using a single device in children as well as their parent who opted to pre-test the sampling procedure before their child. These studies were approved by the Sainte-Justine UHC Human Research Ethics Committee (approval #2021–3067). Participants provided written informed consent (and assent for children) for study participation.

The implementation phase was conducted in two (adult and pediatric) randomized clinical trials in participants who chose remote blood testing and research visits instead of in-person appointments. Investigation Testing Authorization (ITA) applications were approved for the use of the Tasso-SST device in the (i) adult trial entitled *PRevention of COVID-19 With Oral Vitamin D Supplemental Therapy in Essential healthCare Teams (PROTECT) trial*—ITA #322424 (February 8 to May 4, 2021)—NCT04483635; [13] (ii) the pediatric trial entitled *Vitamin D In the Prevention of Viral-induced Asthma in Preschoolers (DIVA) trial*—ITA #334647 (January 27 to February 17, 2022)—NCT03365687 [15]. All sites received ethics approval before the beginning of recruitment at their site. Adults or parents of participating preschoolers provided informed consent; children old enough to understand, provided assent. The results of blood analyses are reported elsewhere [16].

### Investigational phase

**Subjects.**   Individuals aged 12 years or older were eligible to participate in the 'adult' study during the investigational phase. Children aged 1 to 17 years (and one of their parents) were eligible for the child-parent dyad study. As this investigational phase was conducted before the widespread availability of testing and COVID-19 vaccination, patients at risk (e.g., recent travel or high-risk contact) or symptomatic (suspected or confirmed) from COVID-19 infection were excluded. Participants received a $10 gift card and were offered their results for the capillary blood Nadal® COVID-19 IgG/IgM Rapid Test (nal von minden GmbH, Moers, Germany) for COVID-19.

**Tested device.**   The TASSO-SST device comprises a collection unit and reservoir as well as packaging material to return the sample. The device is activated through the press of a button, which initiates a retractable lancet (16-gauge) that punctures the skin, facilitated by a slight vacuum (40 kPa); the capillary blood is collected in a small detachable reservoir containing a thixotropic serum separator gel. When the blood flow ceases, the device is peeled off the skin [17]. The TASSO-SST kit includes alcohol wipes, a single-use blood collection device, a collection tube lid, a biohazard bag with a gauge, a band-aid for post-procedure care, and a shipping

box meeting package requirements for UN3373, ensuring safe shipment of biological substances.

**Randomisation and allocation.** In the adult investigational phase, puncture sites (upper arm and lower back) and topical analgesic (yes or no) were randomly assigned using a 4 x 2 factorial design with a block randomization method using SAS PROC PLAN (SAS Institute Software, Cary, NC, USA). The allocation was implemented using sequentially numbered opaque sealed envelopes.

**Outcomes.** In both studies, blood volume and perceived pain were the two co-primary outcomes, the former to plan the number of devices needed for any target blood volume, the latter as an element of acceptability. The amount of capillary blood collected in the detachable collection reservoir was estimated by comparing its volume to a reference reservoir graded in 50 µl increments (maximum: 750 µl); the volume was recorded as the middle value if the measurement was between two marks. The pain was assessed on the 11-point validated Louisiana Pain Scale from 0 (none) to 10 (unspeakable) [18, 19]. Secondary outcomes included adverse health events, appreciation of the training (instructional brochure, instructional video, life guidance by research personnel, and readiness to use the device), acceptability (overall satisfaction, willingness to use again, preference over venipuncture in a hospital, pain of venipuncture), and pragmatic considerations (i.e., sampling duration, the number of devices used, and device failure rate) (Fig 1). In addition, the perceived usefulness of distraction during blood collection was recorded for children in the child-parent dyad. Adverse health events (AHE) were documented immediately and systematically solicited 24 hours later. A questionnaire served to ascertain acceptability among participants, specifically regarding their satisfaction, willingness to use the device again, preference for it over venipuncture in a hospital setting, and the level of pain experienced during the procedure (S1 Table); responses were recorded on a 5-point Likert scale (1: strongly disagree; 5: strongly agree) or as a yes/no answer (S2 Table). The device failure rate was determined by the inability to collect a minimal blood volume of 100 µl in the aliquot tube, translating into approximately 50 µl of serum, the minimum volume for many analytes (such as calcium/phosphorus/alkaline phosphatase or 25-hydroxyvitamin D). The "sampling duration" refers to the time elapsed from pushing the button on the device and the cessation of blood collection.

**Procedures.** Before sampling, subjects were instructed to watch the online instructional video (2.25 minutes for adults: 8.75 minutes for children and parents) and read the Health Canada-approved study- and age-specific step-by-step graphic brochure on using the Tasso-SST device (S1 Fig). Before using the device, the designated site was vigorously rubbed for 45 seconds and disinfected. In the adult investigational study, approximately 5 grams of 4% lidocaine (Maxilene 4 cream, RGR Pharma Ltd., Lasalle, ON) covered by waterproof transparent dressing (Tegaderm, 3M Canada, London, ON) was applied 30 minutes before capillary sampling for the two sites allocated to topical analgesia; a blood sample of 6–10 ml was also collected via venipuncture by a qualified nurse. The same training and capillary site preparation process were followed in the child-parent dyad study with the following differences: (i) parents were offered to pre-test one sampling device on their arm (and be included as study participants) before using it on their child, (ii) only one device was applied per child either on the upper arm (≥4 years) or lower back (<4 years) based on age; (iii) no topical anesthesia or venipuncture was offered.

## Implementation phase

**Subjects.** The study selection of the two (adult and pediatric) Phase 3 interventional trials, each testing high-dose vitamin D supplementation, has been described elsewhere [11, 13].

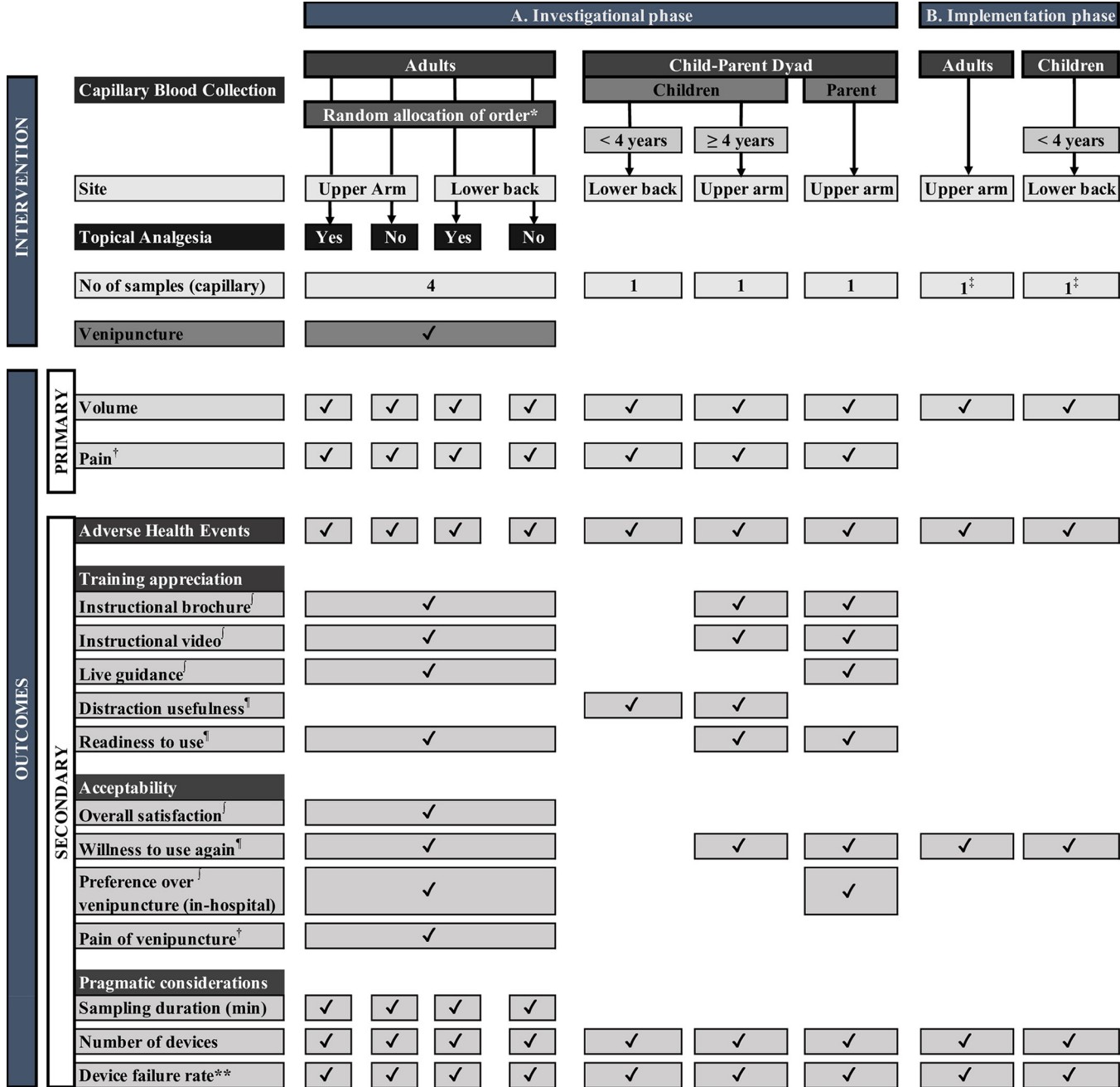

*During the adult investigation phase, 4 capillary blood samples were collected at different sites (arm or back), with/without topical anesthetic; an additional venous sampling was added for each participant.

† Pain was recorded on a 10-point likert scale (0: no pain, 10: unspeakable pain), using the Lousiana Pain Scale.

‡ To meet the targeted blood volume, multiple devices (4 for adults and 2 for children) were provided during implementation

∫ Outcomes recorded on a 5-point likert scale (0: Strongly disagree; 5: Strongly agree).

¶ Outcomes recorded as a Yes or No

** Device failure rate is determined by the ability to collect sufficient volume

**Fig 1. Schedule of procedures testing the applicability and safety of using self-capillary blood collection.**

Briefly, healthcare professionals enrolled in the PROTECT trial and preschoolers (aged 1 to 5 years) enrolled in the DIVA trial were given the option of an in-person visit with a venipuncture or a remote visit with capillary blood drawn via the Tasso-SST device during each encounter requiring blood sampling. In the PROTECT trial, capillary blood collections were done remotely, whereas, in DIVA, capillary blood was collected either remotely or on-site for preschoolers who were unable or refused to have venipuncture. Each adult was given four devices, while each child received up to two devices to ensure the collection of the desired blood volume.

**Outcomes.** The main outcome was the blood volume collected. Secondary outcomes included adverse health events (reported immediately or spontaneously later), acceptability (willingness to use the device again), and pragmatic consideration (number of devices, failure rate, and sampling problems) (Fig 1).

### Statistical analysis

Descriptive statistics served to report participant demographics and outcomes. After the Shapiro-Wilk normality test to verify the continuous outcomes distribution, data were presented as mean ± standard deviation [SD] or median (25%, 75%), as indicated. We conducted a two-way analysis of covariance (ANCOVA) to explore the impact of topical analgesia and puncture site on the co-primary outcomes of pain and volume, with age (continuous), sex (dichotomous), order, and hand dominance as potential covariates. When a significant main effect or interaction was found, pairwise comparisons described the observation. When analyzing data, we considered log-transformed Likert scales to improve the normal distribution. Robust mixed ANOVA was performed when the data did not completely meet the assumptions underlying the analysis of variance [20–22]. Patient-reported outcomes measured on 5-point Likert scales were presented as diverging stacked bar charts.

As a *post hoc* analysis, we investigated the impact of age (continuous), sex, and body mass index (BMI) categorized as per the World Health Organization (WHO) classification, on capillary blood volume across all phases using multiple linear regression analysis [23]. To ensure uniformity and avoid over-representing individuals with multiple capillary samples, we considered only the volume obtained with the first Tasso-SST device applied on the upper arm (except for children <4 years) without topical anesthesia. A case-wise deletion approach was performed for the primary analysis because of the low of missing data (due to participant withdrawal or device failure) [24]. Statistical analyses were conducted using R version 4.2.1 (R Foundation for Statistical Computing, Vienna, Austria, 2022) and STATA version 15 (Stata Corp., College Station, Texas). All tests were 2-sided; an alpha level <0.05 was deemed statistically significant.

## Results

Ninety individuals participated in the adult study, with one person withdrawing consent; 9 children (2 to 17 years) and 7 parents participated in the "child-parent dyad" study. During the implementation phase, 15 (44%) of 34 adult healthcare professionals enrolled in the PROTECT trial and during the 4-month test period of the DIVA trial, 2 (4%) of 47 preschoolers participated in at least one capillary blood collection (Table 1).

### Investigational phase

In the adult study, the median (25%, 75%) capillary blood volume collected, irrespective of puncture site (back or arm) and topical analgesia use (or not), was 450 (250, 550) μl; no statistically significant difference was observed by, or interaction between, puncture sites and analgesia use (Fig 2A). Females, on average, collected 41 μl more blood than males after controlling

**Table 1. Characteristics of participants.**

| | A. Investigational phase | | | B. Implementation phase | |
|---|---|---|---|---|---|
| | Adult | Child | | PROTECT | DIVA |
| | N = 90 adults | N = 9 children | N = 7 parents | N = 15 adults | N = 2 children |
| **Age** | | | | | |
| Median (25%, 75%) | 32 (26, 40) | 14 (12, 16) | 44 (43, 45) | 48.0 (25.0, 58.0) | 2 (1.5, 2) |
| **Male, n (%)** | 38 (44.1) | 5 (55.6) | 2 (28.6) | 0 (0) | 1 (50) |
| **Body Mass Index*** | (n = 25)† | | | | |
| Normal weight, n (%) | 19 (76) | 5 (55.6) | 3 (42.9) | 9 (60) | 2 (100) |
| Overweight, n (%) | 5 (20) | 2 (22.2) | 2 (28.6) | 2 (13.33) | 0 (0) |
| Obese, n (%) | 1(4) | 0 (0) | 2 (28.6) | 4 (26.66) | 0 (0) |
| **Hand dominance** | | | | | |
| Right, n (%) | 83 (92.2) | n/a | n/a | n/a | n/a |
| Left, n (%) | 7 (7.78) | n/a | n/a | n/a | n/a |

*To determine weight categories, we used the World Health Organization BMI-for-age Z-scores for participants aged 1–19 years. Those with scores from -2 to 0.99 were classified as underweight or normal weight, 1–1.99 as overweight, and 2–2.99 as obese. For adults over 19 years, we categorized individuals based on their absolute BMI: >18.5 to <25 as normal weight, >25 to <30 as overweight, and >30 as obese.

†Of note, the height and weight required for calculating Body Mass Index (BMI) were recorded for only 25 out of 90 participants of the investigational adult study when we began to suspect that blood volume might be influenced by obesity, then it was collected for all successive phases.

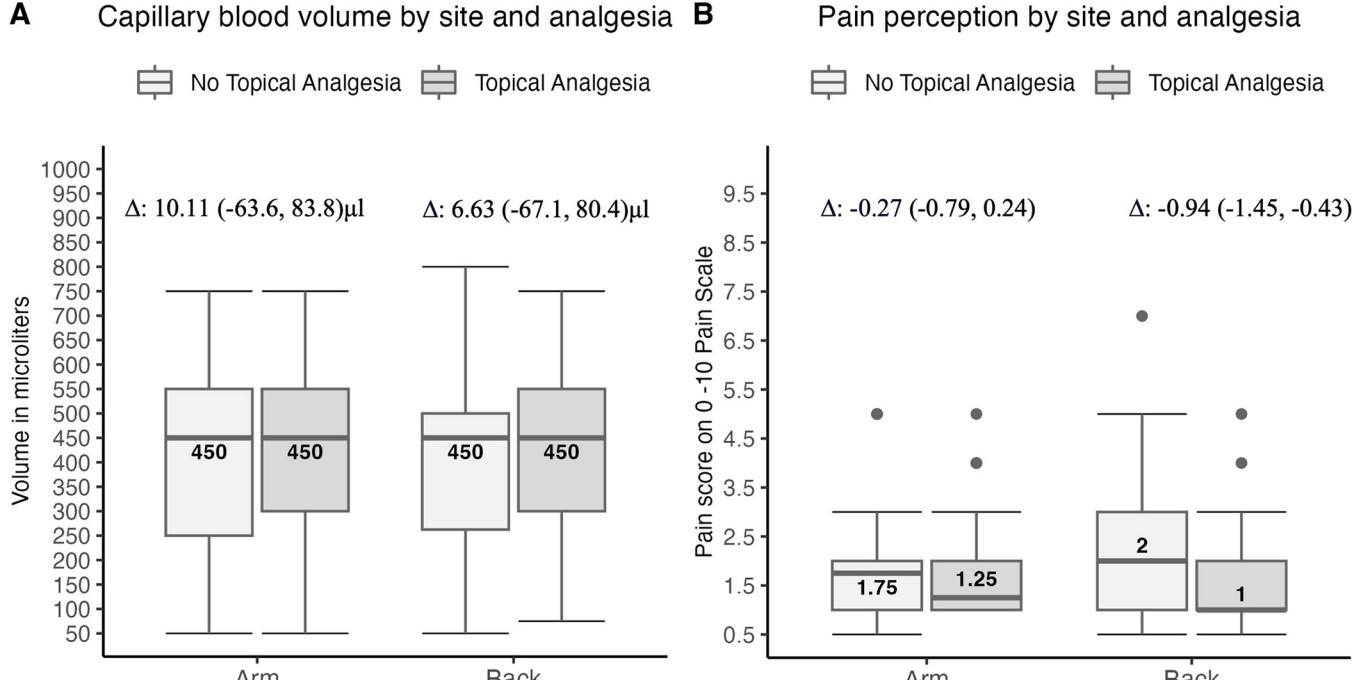

**Fig 2. Distribution of blood volume and pain perception in the adult investigational phase.** The box plots show the distribution of median capillary blood volume (in μl) in A and of the pain level on the 11-point Louisiane pain scale in B, displayed by puncture site (upper arm or lower back) and use of topical analgesia (use in dark or no in clear boxes). The median volume is depicted in each box. The mean group difference with 95% CI within the puncture sites, between topical analgesia use or not, is displayed on top of the box plots.

for puncture site, analgesia use, and age (continuous) ($P = 0.03$). The median perceived pain without topical analgesia was 1 (1, 2) points. Topical analgesia significantly reduced the pain by 0.61 (95% CI: 0.24, 0.97; $P <0.001$) points on the 11-point scale, with a significant interaction observed between puncture site and analgesia use; perceived pain was higher by an average of 0.94 (95% CI: 1.5, 0.4) point for a puncture on the lower back without, compared to the arm with, analgesic (1.87 vs. 0.93; $P<0.001$) (Fig 2B). Of note, the order of capillary samplings and dominant-hand side did not significantly affect blood volume or pain. A Mann-Whitney test revealed that venipuncture was perceived as significantly more painful than the capillary pain score at two sites (arm and back) without topical analgesia (2(1, 3) vs.1(1, 2); $z = -3.367$, $P<0.001$) (S2 Fig).

In the "child-parent dyad" study, the median capillary blood volume collected in 9 children and 7 adults was 450 (400, 475) µl and 350 (200, 400) µl, respectively; no statistically significant difference was observed across age groups on this small sample. In children aged 4 years and over, the median (range) pain score was 0.5 (0 to 7); the parent of the 2-year-old reported the child's pain at 4. Participating parents who tested the device on themselves reported minimal median (range) pain (0 [0 to 1] point) (S3 Fig).

After watching the instructional video, most participants also appreciated the brochure and recommended that both resources be provided for training; all participants expressed readiness to use the Tasso-SST device. Most participants (83% of adults) found live guidance helpful (Fig 3). The majority of the adults (96%) and parents (88%) reported positive experiences with the capillary self-collection, and most (88% adults and 67% parents) preferred the self-collection device over a venipuncture at a hospital (S3 Table). The median sampling duration per device during the investigational phase (adult and child-parent dyad) was 5.01 (3.33, 6.38) minutes.

## Implementation phase

The 15 adults from the PROTECT trial provided 29 capillary samples, each device collecting a median of 350 (250, 450) µl; the 2 children in the DIVA trial provided 4 capillary samples, each device collecting 200 (225, 275) µl (Fig 4). Five adults (17.2%) encountered issues with at least one sampling, namely, insufficient blood and blood coagulation; no such problems were reported for children. No adverse health events were reported.

## Combined investigation and implementation phases

Combining all 123 participants, a median of 425 (250, 500) µl was collected with the first device used on the age-recommended puncture site without topical analgesia. In the 59 participants in whom BMI was documented, no statistically significant impact of age (<4 vs. ≥4 years), sex, and BMI category (normal, overweight, obese) was observed. However, across all study phases (435 successful samplings), a median of 450 (250, 500) µl of capillary blood was collected per device.

Of the 374 successful capillary blood draws in the investigational phase, 11 (12.2%) participants reported adverse effects immediately or within 24 hours; all were minor. Of note, two self-resolving vasovagal syncopes occurred in an adolescent, one during the venous and one during the first capillary sampling (Table 2). In all phases of the study, participants and research staff expressed willingness to use the device again. The overall failure rate was 4.4% (20/452), attributed to device malfunction, blood coagulation, or insufficient blood collection (<100 µl).

## Discussion

Our research evaluated the real-life effectiveness, acceptability, and safety of capillary self-collection using the Tasso-SST device in adults and children. Across all phases of the study, a

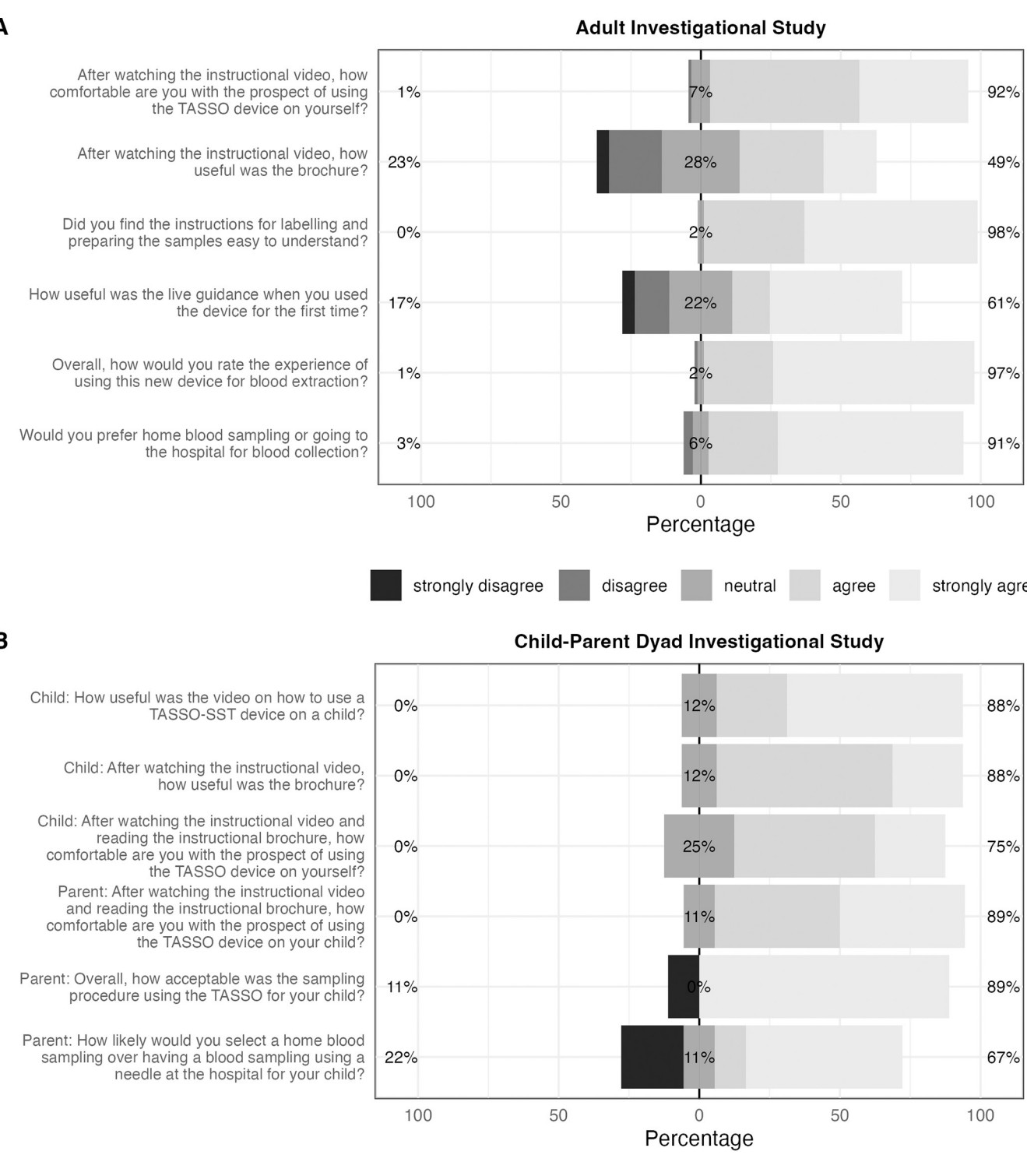

**Fig 3. Feedback from participants on training, satisfaction and preference with the capillary sampling device.** Questionnaire responses of participants enrolled in the adult investigation phase are displayed in Panel A, and those of children and their parent enrolled in the child-parent dyad in Panel B. A bar chart is provided for each question on the y-axis regarding the instructional materials, live guidance, experience and preference. The proportion of participants with agreement (i.e., answered 4 or 5 on the 5-point Likert scale) is displayed on the right of the identity line. In contrast, neutral responses are straddling, and disagreement is shown on the left of the identity line.

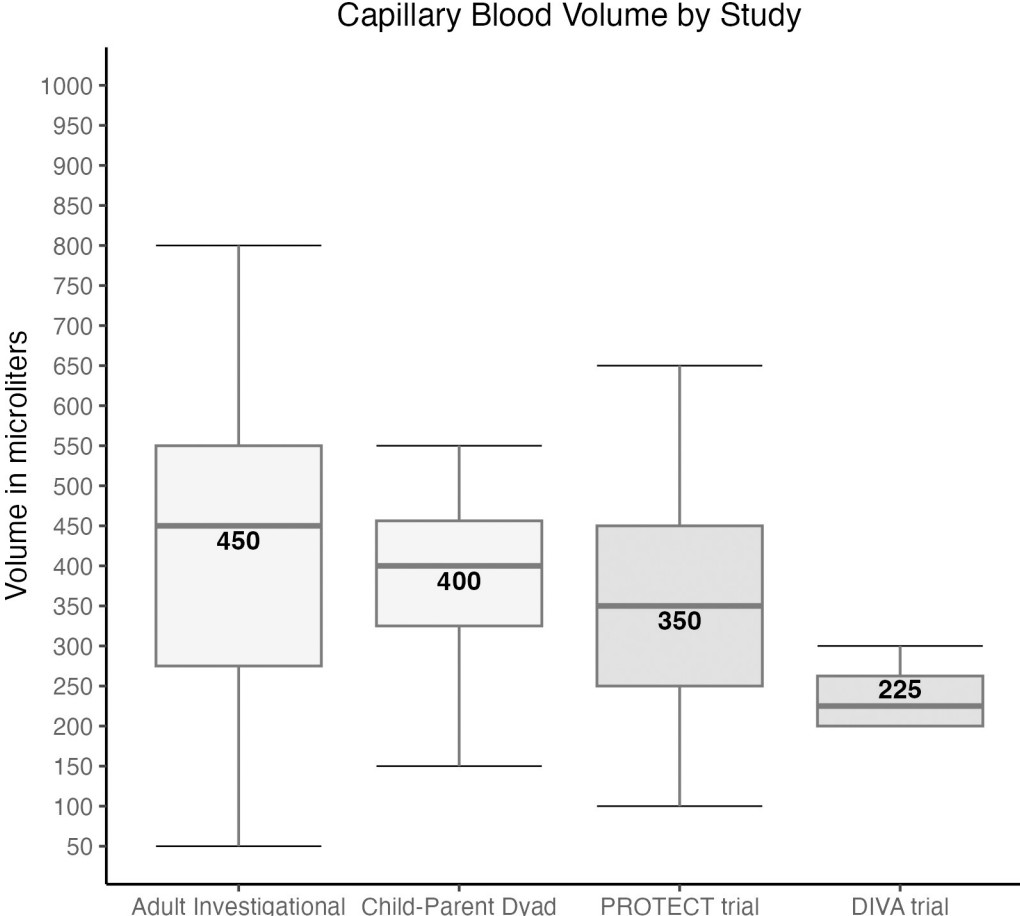

**Fig 4. Distribution of capillary blood volume per device across all phases.** The box plots show the distribution of median capillary blood volume (in μl) per device in the adult and child-parent dyad study during the investigational phase and in the PROTECT and DIVA studies during the implementation phase. Horizontal lines within each box plot represent the median.

median of 450 μl of capillary blood was collected per device. Minimal pain was reported during the investigational phase. Overall, there was great appreciation of the training by video and brochure, high satisfaction with the device, low failure rate, and few adverse health effects.

With a 45-second vigorous rubbing period, we collected a median of 150 to 250 μl more capillary volume than the 200–300 μl range advertised in the manufacturer's brochure [25]. Our median volume of 450 μl was larger than in other studies using the same device [4, 9]. Of note, the puncture site, use of topical analgesia, or dominant side did not significantly affect blood collection volume. When adjusted for confounders, including age, females tended to collect 41 μl more capillary blood than males in the adult investigational study; this effect disappears when combining all phases. Obesity did not significantly impact blood volume, contrary to our initial suspicion that it could decrease capillary blood flow [26]. Consequently, we suspect that site preparation with prolonged vigorous rubbing before puncture may have been the most determinant factor in maximizing capillary blood volume.

When deciding how many devices to provide for blood sampling at home, device failure and target blood volume must be considered. In our experience, the failure rate for collecting a minimum of 100 μl of blood was less than 5%. Other studies using the Tasso-SST device reported failure rates (either insufficient blood or inability to collect any blood) ranging from

Table 2. Adverse health events.

| | A. Investigational phase | | | | B. Implementation phase | |
| | Adult | | Child-Parent Dyad | | PROTECT | DIVA |
| | N = 90 | N = 90 | N = 7 adults | N = 9 children | N = 15 adults | N = 2 Children |
| Sampling method | Venous | Capillary | Capillary | Capillary | Capillary | Capillary |
| Subjects with ≥ 1 adverse events, n (%)* | 11 (12.2) | 11 (12.2) | 0 | 0 | 0 | 0 |
| Number of sampling | n = 90 | n = 358 | | | | |
| Immediate† | | | | | | |
| Vasovagal syncope | 1 (1.1) | 1 (0.3) | 0 | 0 | 0 | 0 |
| Bleeding ‡ | 0 | 1 (0.3) | 0 | 0 | 0 | 0 |
| In the next 24 hrs† | | | | | | |
| Bruise | 9 (10) | 6 (1.7) | 0 | 0 | 0 | 0 |
| Itchiness | 0 | 1 (0.3) | 0 | 0 | 0 | 0 |
| Swelling | 1 (1.1) | 1 (0.3) | 0 | 0 | 0 | 0 |
| Tenderness | 5 (5.6) | 7 (2.0) | 0 | 0 | 0 | 0 |
| Total† | 16 (17.8) | 16 (4.5) | 0 | 0 | 0 | 0 |

*Number of subjects reporting an AE on at least one puncture site of the TASSO application and venipuncture.

† Number of immediate and 24-hour adverse events are reported as a percentage of total number of (venipuncture and capillary) samplings. Delayed AHE may be underreported during the implementation phase as AHE in the next 24 hours was actively sought for the investigational phase.

‡Indicates prolonged (more than 5 minutes) bleeding after blood collection at the puncture site.

3.3% to 20% [4, 8, 9, 27, 28]. To achieve the desired blood volume, we recommend providing a sufficient number of devices, calculated by assuming no more than a median blood volume of 450 µl per device, with extra devices to accommodate suboptimal volume (250 uL (lower volume quartile) and device failure (5%). For preschoolers, conservatively assuming a volume of 250µl/device may be preferable, based on very limited data.

Minimal capillary sampling pain was reported across the investigational phase with no significant impact of age, gender, dominant side, and BMI. There was a statistically significant difference in pain perception between puncture sites. Adolescents and adults reported a median pain of 2 points for puncture in the lower back without analgesia compared to 1 point for all other site-analgesia combinations. Given the small sample size and absence of a factorial design, we could not validate these findings and effect size among children, let alone preschoolers for whom the lower back is recommended based on an unpublished focus group. Pain research suggested that the minimal clinically significant difference (MCSD) in children varies between 1 or 2 points; [8, 29, 30], whereas for adults, it was 1.4 points [31]. It is unclear if findings about topical analgesia obtained in individuals 12 years and older can be generalized to children; if so, the observed 0.6-point difference with or without topical analgesia, regardless of site, lacks clinical importance. The extension of these findings to preschoolers below 4 years old is also uncertain. In contrast, adults reported significantly less pain with capillary sampling than with venous sampling, consistent with previous research findings [9, 28]. Therefore, Capillary sampling is generally deemed less painful than venipuncture in adults, rendering it acceptable across all age groups. For very young children, considering the upper arm or providing topical analgesia for the lower back seems reasonable.

Consistent with other studies, [32–34] the Tasso-SST device appears safe with minimal risk of adverse effects. We observed a lower risk of bruising (1.7%) with capillary than venous sampling (10%); our proportion of bruising was lower than reported in similar studies (5%) [32, 34]. In a previous study in adults, 8.3% of participants exhibited scarring at the puncture site 90 days post-blood collection [28]. However, we did not systematically solicit adverse effects

beyond 24 hours in our study. Of note, as one participant experienced a vasovagal syncope with both venous and capillary samplings, it seems reasonable to systematically enquire about such predisposition; if present, we advise caution and ensuring to have an accompanying adult to prevent any fall during the capillary sampling [34].

The technical and logistical challenges of implementing a completely remote blood collection require clear instructions describing the procedure and shipping instructions [34, 35]. After watching our instructional video and reading the brochure, all participants felt comfortable testing the device during the investigation phase. These findings were similar to the Human Epidemiology and RespOnse to the SARS-CoV-2 (HEROS) study, in which 87% of participants were confident in testing the Tasso-SST device after reviewing the provided instructional materials [34]. Most participants in our study found the packaging and shipping instructions easy to understand. Live guidance was helpful for many adults during first use, and some parents expressed interest in receiving it again when using the device with their child.

In line with recent studies, [9, 33, 36–38]. an overwhelming majority of participants had a positive experience with the device for capillary self-sampling. Still, some parents preferred venous sampling, perhaps due to discomfort about being responsible for the procedure instead of medical personnel. Overall, all participants expressed their willingness to use the device again, consistent with previous studies conducted on adults and children [33, 34, 36, 38]. The median duration of sampling using the device was comparable to a prior study that employed the TASSO-SST device [8].

We acknowledge the following strengths and limitations. Our rigorous design to test the impact of the puncture site and topical analgesia use on blood volume and pain perception in adolescents and adults, confirming no impact on the volume and negligible impact on pain, may not be generalizable to young children. We did not repeat the factorial design in children to avoid subjecting them to four punctures; caution is advised when extrapolating study results to young children. Yet, our estimate of volume, acceptability, pragmatic considerations, and safety are based on a total of 119 participants of all ages, with a low dropout rate. Adverse events were documented for 24 hours in the investigational phase and throughout the study participation, which lasted 4 to 8 weeks in the PROTECT trial and 7 months in the DIVA trial. There were no reports of long-term adverse effects or scarring, but long-term local effects were not systematically checked. During the pandemic, when we offered remote trial visits (and blood sampling), we did not investigate the reason(s) for preferring in-person visits, nor could we distinguish factors related to the visits (convenience of the visit place and time, other elements of the medical/research follow-up), sampling (comfort with procedure, flexibility of time) and host (health care professionals in PROTECT vs. parents of very young children in DIVA) affecting this decision. The cohort selection in the implementation phase, particularly health care professionals in PROTECT, may have resulted in an overestimation of observed feasibility and satisfaction. We suspect the low uptake of remote visits in the DIVA trial may be due to a preference for in-person medical visits and hesitation to use the technique on young children. More data is needed to evaluate implementation in preschoolers.

## Conclusion

In summary, capillary blood collection using the Tasso-SST device appears to be an acceptable, safe, and relatively painless method with high satisfaction that can serve as a viable alternative to venous blood collection in adults and children. With good guidance, self-collected capillary blood devices can be used to conduct clinical trials remotely and as an alternative to venipuncture on-site in participants in all age groups.

## Supporting information

**S1 Table. Questionnaire ascertaining training appreciation and acceptability in the adult study of the investigational phase.** Reprinted from [25] under a CC BY license, with permission from Tasso Inc., original copyright [2020].
(DOCX)

**S2 Table. Questionnaire ascertaining training appreciation and acceptability in the child-parent dyad study of the investigational phase.**
(DOCX)

**S3 Table. Patient-reported outcomes.**
(DOCX)

**S1 Fig. Instructional brochure on the application of TASSO-SST device provided to participants in the investigational phase.**
(PDF)

**S2 Fig. Reported pain by puncture site and topical analgesia use in the adult investigational phase.**
(PDF)

**S3 Fig. Reported pain by age group in the child-parent dyad investigational phase as above.**
(PDF)

## Acknowledgments

We acknowledge the infrastructure support of the Fonds de la Recherche du Québec en Santé (FRQS) provided to the Research Institutes of the Sainte-Justine University Health Centre (SJUHC). We thank the research staff and study participants enrolled in all study aspects.

## Author Contributions

**Conceptualization:** Francine M. Ducharme.

**Data curation:** Harika Dasari, Anna Smyrnova.

**Formal analysis:** Harika Dasari.

**Funding acquisition:** Francine M. Ducharme.

**Investigation:** Harika Dasari, Francine M. Ducharme.

**Methodology:** Anna Smyrnova, Jing Leng.

**Project administration:** Jing Leng.

**Software:** Anna Smyrnova.

**Supervision:** Anna Smyrnova, Francine M. Ducharme.

**Validation:** Harika Dasari, Anna Smyrnova, Francine M. Ducharme.

**Visualization:** Harika Dasari, Anna Smyrnova, Francine M. Ducharme.

**Writing – original draft:** Harika Dasari.

**Writing – review & editing:** Anna Smyrnova, Jing Leng, Francine M. Ducharme.

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
