## [Decision Letter · Decision Letter 0]

6 Dec 2023

PONE-D-23-28982Feasibility, acceptability, and safety of a novel device for self-collecting capillary blood samples in clinical trials in the context of the pandemic and beyondPLOS ONE

Dear Dr. Ducharme,

Thank you for submitting your manuscript to PLOS ONE. After careful consideration, we feel that it has merit but does not fully meet PLOS ONE’s publication criteria as it currently stands. Therefore, we invite you to submit a revised version of the manuscript that addresses the points raised during the review process.

it a nice work .. but you need to check the English language and submit the row data 

The device exists, is in use, and has been tested. However, I did not understand the desired benefit from the research

first the sample size are really too small to drew a conclusion 

the pain issue can't be measure just by asking ... also it is not the main issue when you collecting blood

I really like the idea of using the device ... mainly maybe for cancer patients 

We look forward to receiving your revised manuscript.

Kind regards,

Ramada Rateb Khasawneh

Academic Editor

PLOS ONE

Journal Requirements:

2. Please ensure you have included the registration number for the clinical trial referenced in the manuscript.

"FMD has received unrestricted research funds from AstraZeneca, Covis Pharma,

GlaxoSmithKline, Merck Canada, Novartis, Teva, Trudell Medical; GlaxoSmithKline

and MEDteq in partnership with Thorasys Inc., as well as an honorarium for

consultancy work from AstraZeneca, Covis Pharma, Sanofi, Teva, and Thorasys Inc.;

and honorarium as an invited speaker from Covis Pharma, Jean-Coutu Pharmacy and Brunet Pharmacy. All other co-authors have no conflict of interest."

7. We note that Figure S1 in your submission contain copyrighted images. All PLOS content is published under the Creative Commons Attribution License (CC BY 4.0), which means that the manuscript, images, and Supporting Information files will be freely available online, and any third party is permitted to access, download, copy, distribute, and use these materials in any way, even commercially, with proper attribution. For more information, see our copyright guidelines: http://journals.plos.org/plosone/s/licenses-and-copyright.

a. You may seek permission from the original copyright holder of Figure S1 to publish the content specifically under the CC BY 4.0 license. 

Additional Editor Comments:

it is a nice paper

but you need to check the English language and submit the row data

Reviewers' comments:

Reviewer's Responses to Questions

**Comments to the Author**

1. Is the manuscript technically sound, and do the data support the conclusions?

Reviewer #1: Yes

Reviewer #2: Yes

2. Has the statistical analysis been performed appropriately and rigorously? 

Reviewer #1: Yes

Reviewer #2: I Don't Know

3. Have the authors made all data underlying the findings in their manuscript fully available?

Reviewer #1: Yes

Reviewer #2: Yes

4. Is the manuscript presented in an intelligible fashion and written in standard English?

Reviewer #1: Yes

Reviewer #2: Yes

5. Review Comments to the Author

Reviewer #1: Paper Review: "Feasibility, Acceptability, and Safety of a New Device for Self-Collection of Capillary Blood Samples in Clinical Research in Pandemic and Post-Pandemic Situations" Abstract: This paper shows that gender, acceptability, and safety of new self-collection capillary blood devices, particularly the Tasso SST device.

The study consists of two phases: a research phase with on-site visits and an implementation phase with remote visits.

Results will focus on blood volume, pain perception, device failure, adverse events, and participant satisfaction.

Advantages:

1.Comprehensive approach: Comprehensive design of the study covering both on-site and remote scenarios provides a comprehensive understanding of the device's applicability.

2.Different participant demographics: The inclusion of adults and children in the study phase with different puncture sites and analgesic options increases the relevance of the study and increases the potential for broader application .

3.Clear display of results: Clear display of results such as mean blood volume, change in pain perception, device failure rate, and adverse events. This transparency helps interpret the meaning of the research.

4.Practical Implementation: The inclusion of a remote implementation phase is in line with the evolving trend of decentralized healthcare and increases the practical value of the research.

Areas for improvement:

1. Long-term follow-up: Although this article refers to safety and satisfaction, longer-term follow-up may improve continued adoption and safety of self-collection devices over time. It may give you some insight about sexuality.

2. Comparative analysis: Comparative analysis using traditional venipuncture or other self-sampling devices strengthens the paper's contribution by providing context and benchmarks.

3.Variation in pain perception: Examining variation in pain perception across different individuals and age groups can lead to a more nuanced understanding of subjective experience.

4.Discussion of Limitations: Although the results are promising, a discussion of the study's limitations, such as potential biases or specific challenges encountered during the study, would add additional depth to the paper.

Conclusion: This article provides valuable insight into the feasibility, acceptability, and safety of the Tasso SST device for self-collection of capillary blood.

This study contributes to the evolution of the landscape of remote monitoring in clinical trials with its integrative design and clear presentation of results.

Implementation of the suggested improvements will further increase the impact and relevance of the article in the broader medical field.

Reviewer #2: The article explores the feasibility of capillary blood self-collection using the Tasso-SST device for remote monitoring. I appreciate the authors' attention to this important topic and their thorough exploration of the impact of sampling sites and topical analgesia on capillary blood volume and pain perception. The study's findings have significant implications for remote monitoring and the use of self-collection devices in healthcare, and the high satisfaction and safety levels reported are promising for the future implementation of such methods. While the article is well-written, it is important to ensure that the reference number comes after the punctuation for consistency and clarity in scientific writing.

6. PLOS authors have the option to publish the peer review history of their article (what does this mean?). If published, this will include your full peer review and any attached files.

Reviewer #1: No

Reviewer #2: No

---

## [Author Response · Author response to Decision Letter 0]

29 Feb 2024

Feasibility, acceptability, and safety of a novel device for self-collecting capillary blood samples in clinical trials in the context of the pandemic and beyond [PONE-D-23-28982]

A point-by-point response to the Editors’ and Reviewers’ comments

Please note that our responses hereafter are in bold.

We verified the style templates and updated the manuscript accordingly. 

2. Please ensure you have included the registration number for the clinical trial referenced in the manuscript.

We have included the NCT registration numbers for the PROTECT and DIVA clinical trials accordingly.

We have discussed the issue with the Research Ethics Board; as we did not include a data sharing section in the consent forms, the REB recommends that we release the anonymized data upon request after review of the intended use. We are considering making the final available upon request on the Harvard Data verse: a DOI will be provided upon final acceptance of the manuscript. We have therefore clarified this in the Availability statement section.

"FMD has received unrestricted research funds from AstraZeneca, Covis Pharma,

GlaxoSmithKline, Merck Canada, Novartis, Teva, Trudell Medical; GlaxoSmithKline

and MEDteq in partnership with Thorasys Inc., as well as an honorarium for

consultancy work from AstraZeneca, Covis Pharma, Sanofi, Teva, and Thorasys Inc.;

and honorarium as an invited speaker from Covis Pharma, Jean-Coutu Pharmacy and Brunet Pharmacy. All other co-authors have no conflict of interest."

We confirm that the research funds awarded to FMD do not alter our adherence to all PLOS ONE policies and have included the following statement: "This does not alter our adherence to PLOS ONE policy on sharing data and materials "in the revised manuscript.

An updated competing interests statement has been included in the cover letter.

As addressed in question 3, we have discussed the issue with the Research Ethics Board; as we did not include a data sharing section in the consent forms, the REB recommends that we release the anonymized data upon request after review of the intended use. We are considering the use of the Harvard Data verse but are currently identifying the contact person that would be responsible for data access. In a first step, the authors’ names will be the main contact until an institutional contact/email address devoted to reviewing data access request is decided. A DOI will be provided upon final acceptance of the manuscript. We have, therefore, clarified this in the Availability statement section.

ORCID iD for the corresponding author, Dr. Francine Ducharme was already provided in our initial submission. 

7. We note that Figure S1 in your submission contains copyrighted images. All PLOS content is published under the Creative Commons Attribution License (CC BY 4.0), which means that the manuscript, images, and Supporting Information files will be freely available online, and any third party is permitted to access, download, copy, distribute, and use these materials in any way, even commercially, with proper attribution. For more information, see our copyright guidelines: http://journals.plos.org/plosone/s/licenses-and-copyright.

a. You may seek permission from the original copyright holder of Figure S1 to publish the content specifically under the CC BY 4.0 license. 

We have attached the content permission form for the said Figure S1, under filename ‘Content_Permission_Tasso_2023-12-20.pdf’ and added the requested caption in the legend of Figure S1 appearing on the last page of the manuscript.

b. If you are unable to obtain permission from the original copyright holder to publish these figures under the CC BY 4.0 license or if the copyright holder’s requirements are incompatible with the CC BY 4.0 license, please either i) remove the figure or ii) supply a replacement figure that complies with the CC BY 4.0 license. Please check the copyright information on all replacement figures and update the figure caption with source information. If applicable, please specify in the figure caption text when a figure is similar but not identical to the original image and is, therefore, for illustrative purposes only.

Not applicable.

8. Please review your reference list to ensure that it is complete and correct. If you have cited papers that have been retracted, please include the rationale for doing so in the manuscript text or remove these references and replace them with relevant current references. Any changes to the reference list should be mentioned in the rebuttal letter that accompanies your revised manuscript. If you need to cite a retracted article, indicate the article’s retracted status in the References list and include a citation and full reference for the retraction notice.

We have made modifications to the reference list, addressing an issue where a reference was duplicated during the verification process. Furthermore, we have included one additional reference in the discussion in response to one of the reviewer’s comments. The revised reference list now comprises 38 entries, with the removal of the duplicated entry and the addition of the new reference.

Reviewers' comments:

Reviewer's Responses to Questions

Comments to the Author

1. Is the manuscript technically sound, and do the data support the conclusions?

Reviewer #1: Yes

Reviewer #2: Yes

Thank you, no modifications needed.

2. Has the statistical analysis been performed appropriately and rigorously? 

Reviewer #1: Yes

Reviewer #2: I Don't Know

Thank you, no modifications are needed.

3. Have the authors made all data underlying the findings in their manuscript fully available?

The PLOS Data policy requires authors to make all data underlying the findings described in their manuscript fully available without restriction, with rare exceptions (please refer to the Data Availability Statement in the manuscript PDF file). The data should be provided as part of the manuscript or its supporting information or deposited in a public repository. For example, in addition to summary statistics, the data points behind means, medians and variance measures should be available. If there are restrictions on publicly sharing data—e.g. participant privacy or use of data from a third party—those must be specified.

Reviewer #1: Yes

Reviewer #2: Yes

Thank you, no modifications needed

4. Is the manuscript presented in an intelligible fashion and written in standard English?

Reviewer #1: Yes

Reviewer #2: Yes

Thank you, no modifications needed

5. Review Comments to the Author

Reviewer #1: Paper Review: "Feasibility, Acceptability, and Safety of a New Device for Self-Collection of Capillary Blood Samples in Clinical Research in Pandemic and Post-Pandemic Situations" Abstract: This paper shows that gender, acceptability, and safety of new self-collection capillary blood devices, particularly the Tasso SST device.

The study consists of two phases: a research phase with on-site visits and an implementation phase with remote visits.

Results will focus on blood volume, pain perception, device failure, adverse events, and participant satisfaction.

Advantages:

1.Comprehensive approach: Comprehensive design of the study covering both on-site and remote scenarios provides a comprehensive understanding of the device's applicability.

2.Different participant demographics: The inclusion of adults and children in the study phase with different puncture sites and analgesic options increases the relevance of the study and increases the potential for broader application .

3.Clear display of results: Clear display of results such as mean blood volume, change in pain perception, device failure rate, and adverse events. This transparency helps interpret the meaning of the research.

4.Practical Implementation: The inclusion of a remote implementation phase is in line with the evolving trend of decentralized healthcare and increases the practical value of the research.

Thank you, for the accurate report of our publication.

Areas for improvement:

1. Long-term follow-up: Although this article refers to safety and satisfaction, longer-term follow-up may improve continued adoption and safety of self-collection devices over time. It may give you some insight about sexuality.

We mentioned these issues in the limitations paragraph.

2. Comparative analysis: Comparative analysis using traditional venipuncture or other self-sampling devices strengthens the paper's contribution by providing context and benchmarks.

Thank you.

3.Variation in pain perception: Examining variation in pain perception across different individuals and age groups can lead to a more nuanced understanding of subjective experience.

We agree.

4.Discussion of Limitations: Although the results are promising, a discussion of the study's limitations, such as potential biases or specific challenges encountered during the study, would add additional depth to the paper.

We added the possibility of selection bias resulting from conducting the adult implementation study in Health care professionals may have overestimated feasibility and satisfaction. We mentioned other challenges.

 Conclusion: This article provides valuable insight into the feasibility, acceptability, and safety of the Tasso SST device for self-collection of capillary blood.

This study contributes to the evolution of the landscape of remote monitoring in clinical trials with its integrative design and clear presentation of results.

Implementation of the suggested improvements will further increase the impact and relevance of the article in the broader medical field.

Thank you.

Reviewer #2: The article explores the feasibility of capillary blood self-collection using the Tasso-SST device for remote monitoring. I appreciate the authors' attention to this important topic and their thorough exploration of the impact of sampling sites and topical analgesia on capillary blood volume and pain perception. The study's findings have significant implications for remote monitoring and the use of self-collection devices in healthcare, and the high satisfaction and safety levels reported are promising for the future implementation of such methods. While the a

---

## [Decision Letter · Decision Letter 1]

8 May 2024

Feasibility, acceptability, and safety of a novel device for self-collecting capillary blood samples in clinical trials in the context of the pandemic and beyond

PONE-D-23-28982R1

Dear Dr. Ducharme,

We’re pleased to inform you that your manuscript has been judged scientifically suitable for publication and will be formally accepted for publication once it meets all outstanding technical requirements.

Kind regards,

Ramada Rateb Khasawneh

Academic Editor

PLOS ONE

Additional Editor Comments (optional):

sorry for the delay

the paper met the plos one criteria

it is looks better now ... good luck

Reviewers' comments:

Reviewer's Responses to Questions

**Comments to the Author**

1. If the authors have adequately addressed your comments raised in a previous round of review and you feel that this manuscript is now acceptable for publication, you may indicate that here to bypass the “Comments to the Author” section, enter your conflict of interest statement in the “Confidential to Editor” section, and submit your "Accept" recommendation.

Reviewer #3: All comments have been addressed

Reviewer #4: All comments have been addressed

Reviewer #5: All comments have been addressed

2. Is the manuscript technically sound, and do the data support the conclusions?

Reviewer #3: Yes

Reviewer #4: Yes

Reviewer #5: Yes

3. Has the statistical analysis been performed appropriately and rigorously? 

Reviewer #3: Yes

Reviewer #4: Yes

Reviewer #5: Yes

4. Have the authors made all data underlying the findings in their manuscript fully available?

Reviewer #3: Yes

Reviewer #4: Yes

Reviewer #5: (No Response)

5. Is the manuscript presented in an intelligible fashion and written in standard English?

Reviewer #3: Yes

Reviewer #4: Yes

Reviewer #5: Yes

6. Review Comments to the Author

Reviewer #3: it is nice that you address all the assign issue, the paper sound better now

Good Luck in your future work

Reviewer #4: In all the manuscripts I have reviewed thus far, this is one of the best written, well thought out and answered all my questions.

Reviewer #5: (No Response)

7. PLOS authors have the option to publish the peer review history of their article (what does this mean?). If published, this will include your full peer review and any attached files.

Reviewer #3: No

Reviewer #4: No

Reviewer #5: No

---

## [Editor Report · Acceptance letter]

17 May 2024

PONE-D-23-28982R1 

PLOS ONE

Dear Dr. Ducharme, 

I'm pleased to inform you that your manuscript has been deemed suitable for publication in PLOS ONE. Congratulations! Your manuscript is now being handed over to our production team.

Kind regards, 

on behalf of

Dr. Ramada Rateb Khasawneh 

Academic Editor

PLOS ONE